# Study on the evolution of the spatial patterns and driving factors of national agricultural cultural heritage in China

**Lifei Xu**[1], **Yuyu Liao**[1,2], **Jun Liu**[1,2]*

**1** College of Tourism, Hubei University, Wuhan, Hubei, China, **2** Tourism Development and Management Research Center of Hubei Key Research Base of Humanities and Social Sciences, Wuhan, Hubei, China

\* magicliu@hubu.edu.cn

**Data Availability Statement:** All relevant data are within the paper and its Supporting Information files. If needed, more information about how the authors obtained the data is provided here: The gross regional product, the gross domestic

## Abstract

The study analyzed the spatial distribution characteristics, evolution rules, and driving factors of 138 China's national agricultural cultural heritage sites from 2013 to 2021 at the overall and regional levels, using kernel density analysis, Centres for standard deviation ellipse analyses, spatial autocorrelation analysis, and geographical detector analysis.The results showed that: ①From an overall perspective, the spatial pattern of China's national agricultural cultural heritage changed greatly from 2013 to 2021, with a highly uneven spatial distribution, gradually showing a distribution pattern of "widely distributed, locally concentrated". The spatial distribution of China's national agricultural cultural heritage is increasingly evident, and the spatial distribution type has evolved from discrete to clustered. The spatial distribution center of gravity shows a shift trajectory of "north-east, then south-east". During the study period, the X axis of the standard deviation ellipse was always greater than the Y axis, but the difference between the X and Y axes was small, indicating that the spatial distribution direction was northeast-southwest, but the directionality was weak. The types of national agricultural cultural heritage are diverse and rich, involving farmland landscapes, composite systems, crop varieties, vegetables and melons, tea, forest fruits, special products, farmland irrigation, and animal breeding, in which forest and fruit class heritage site dominate. ②From a regional perspective, the spatial distribution of China's national agricultural cultural heritage varies greatly, with strong national and regional characteristics. The high-density core area of the national agricultural cultural heritage in 2013 was located in the intersection of Fujian, Zhejiang and Jiangxi. After 2017, the high-density core area moved northward to the Yangtze River Delta region, which was caused by a combination of economic, cultural, and geographical factors. In addition, the agglomeration of the intersection edge area of Guizhou, Hunan and Yunnan provinces has emerged in 2021. In analyzing spatial autocorrelation, drawing on existing relevant research results, the study selected county-level administrative districts as the research unit. The analysis results show that there is a positive spatial correlation between China's national agricultural cultural heritage sites in 2017 and beyond, with the spatial distribution types mainly being LL and LH. During the study period, the number of LL and HH type areas has been increasing, indicating that the positive spatial correlation between China's national agricultural cultural heritage sites is gradually

product of the primary and tertiary industries, general public budget expenditure and per capita regional GDP were obtained from the statistical yearbooks or bulletins of the provinces, cities, and counties. The population data were obtained from spatial demographic statistics provided by Landscan, and some of the missing data were obtained from statistical yearbooks or bulletins of various provinces, cities, and counties.The data of the national key cultural relic protection units came from the national integrated online government affairs platform (the comprehensive administrative management platform of the State Administration of Cultural Heritage). The data of the national A-level scenic spots are from the official websites of the culture and tourism departments of the provinces, autonomous regions, and municipalities directly under the Central Government and the government affairs data resource websites. The precipitation and temperature data are from the ERA5-Land dataset released by the European Union and the European Center for Medium-range Weather Forecast.The elevation data were obtained from the Resources and Environmental Sciences and Data Center of the Chinese Academy of Sciences, and the river and road data were obtained from the National Geographic Information Resource Catalog Service System.

**Funding:** this work was supported by grants from National Social Science Foundation Project "Research on the Accounting, Influencing Factors, and Emission Reduction Pathways of China's Inter provincial Tourism Carbon Emissions" [grant number 23BJY142].Details of the funded research are as follows: Initials of the authors who received each award: J L Grant numbers awarded to each author: [23BJY142] The full name of each funder: Jun Liu The funder provided guidance and support in the study design, data collection and analysis, decision to publish, or preparation of the manuscript.

**Competing interests:** There are no competing interests.

strengthening.③In terms of influencing factors, the spatial pattern of China's national agricultural cultural heritage is affected by factors such as regional economic development level, policy guarantee, transportation accessibility, cultural environment, per capita economic development level, population status, primary industry economic development level, secondary industry economic development level, tourism resource endowment, temperature, precipitation, terrain, and rivers. Among them, the impact of tourism resources, regional economic development level, and policy guarantees are more significant. The explanatory power of the interaction between any two factors is greater than that of a single factor, and there are differences in the strength of the interaction between each influencing factor.

## Introduction

In response to the global trend of destroying family farming and traditional agricultural systems, in 2002, the Food and Agriculture Organization of the United Nations launched a global partnership initiative for the protection and adaptive management of "globally important agricultural cultural heritage systems". The Food and Agriculture Organization of the United Nations defines a globally important agricultural cultural heritage system as "an existing and continuously developing human community system that is in a complex relationship with its location, cultural or agricultural landscape, or biophysical, and wider social environment". Agricultural cultural heritage is a composite heritage that covers the characteristics of tangible heritage, intangible heritage and intangible cultural heritage, as well as a composite social-eco-economic system with significant social, ecological and economic benefits [1, 2]. Agricultural cultural heritage is living, adaptable, dynamic, composite and sustainable [3]. This system and its landscape are rich in biodiversity and can promote the development of local socioeconomic and cultural factors [4]. It is of great value for the sustainable development of the region and the improvement of the rural ecological environment [5]. As a new type of heritage, agricultural heritage is facing the challenges that agricultural models centered on agricultural intensification have weakened traditional agroecosystems, the loss of natural resources caused by unsustainable farming methods, and the deterioration of weather conditions and nature caused by climate change. There are a series of unprecedented challenges, such as changes in the medical cycle [6–8].

As one of the important origins of agriculture in the world, China's long history has given birth to diverse agricultural practices. During long-term interactions with the natural environment, various ethnic groups have developed rich and varied, stable, efficient and sustainable land use patterns according to local conditions [9]. China was one of the first countries to respond to and actively participate in the protection of agricultural heritage. In 2005, the Qingtian rice-fish symbiosis system in Zhejiang Province was identified as China's first globally important agricultural cultural heritage site, marking the official start of agricultural heritage protection in China [10]. In 2012, China formulated the "Criteria for the Identification of China's Important Agricultural Cultural Heritage" to actively explore the management mechanisms and development paths of agricultural cultural heritage [11]. In 2013, China launched the first batch of national agricultural cultural heritage assessment works. As of 2022, China has designated 6 batches of 138 national-level agricultural cultural heritage items. According to the classification by thematic content of the China Agricultural Museum, the national-level agricultural cultural heritage involves farmland landscapes, composite systems, crop varieties,

tea, forest and fruits, special products, farmland irrigation, animal breeding, vegetables and melons. Chinese farming civilization has a profound history, its agricultural culture is rich and diverse, and numerous agricultural cultural heritages still need to be discovered and protected. Therefore, the study of China's national agricultural cultural heritage is particularly urgent.

In recent years, the research content of agricultural cultural heritage has gradually enriched, and the research field has gradually shifted from theory to practical application research. The research on agricultural cultural heritage involves the concept of agricultural cultural heritage [1], protection and management [12], tourism development [13, 14], innovative development [15], responsible behavior [16], and ecological value [17]. Among them, the value, sustainable development, spatial distribution and influencing factors of agricultural cultural heritage are hot topics for scholars to study. For example, Min, Zhang et al. divided the value of agricultural heritage into historical, socio-cultural, ecological, aesthetic, economic and scientific research values [18]. Su divides the multiple values of agricultural heritage systems into existential value, functional value, and strategic value. Existential value includes ecological value, livelihood value, historical value, and socio-cultural and aesthetic value. Functional value includes economic, scientific, and educational value [19]. Sheryl Rose C. Reyes conducted a comprehensive review of 11 GIAHS application proposals from Japan, proposing a set of indicators and perspectives to address challenges and opportunities and promote the sustainable development of agricultural cultural heritage [20]. Lu H constructed a tourism development framework for Globally Important Agricultural Heritage Systems (GIAHS) through Multi-Functional Agriculture (MFA), leveraging the multifunctionality of agriculture to ensure the sustainability of agricultural cultural heritage tourism [21]. With the rise of GIS spatial analysis software, spatial analysis techniques have been widely used to explore the spatial distribution characteristics, patterns, and evolution of research elements in specific region [22]. In spatial analysis tools, all the analysis elements have point characteristics with clear longitude and latitude coordinates. The statistical analysis of the discrete distribution is used to reveal their spatial distribution patterns and characteristics [23]. For example, Liu et al. used spatial analysis tools such as density analysis and buffer analysis to analyze the spatial distribution characteristics of agricultural cultural heritage and investigated coupling with the tourism industry and the degree of response [11]. Han Zongxin used spatial neighbor analysis and spatial autocorrelation methods such as methods and kernel density analysis to analyze the distribution characteristics of Chinese agricultural cultural heritage [24]. Guo Xuan et al. used analytical tools such as mean nearest neighbour analysis, kernel density analysis, and standard deviation ellipse analysis to reveal the spatial distribution characteristics of China's national key agricultural heritage system [25].

Academic research has deepened the theoretical implications of agricultural cultural heritage, expanded the research perspective, and is highly important for future research on agricultural cultural heritage. Chinese scholars began to focus on the research of agricultural cultural heritage in 2006, but to date, there have been relatively few studies on the evolution of spatial patterns and driving factors of agricultural cultural heritage, especially on a national scale. In the existing studies on the spatial pattern of agricultural cultural heritage, most of the emphasis was placed on the static analysis of spatial distribution, and the analysis of the spatiotemporal dynamic evolution of agricultural cultural heritage was lacking. This will limit our in-depth understanding of the evolution patterns and trends of agricultural cultural heritage, and prevent us from providing more comprehensive and scientific support for conservation efforts. In studies on influencing factors, research methods were mainly focused on qualitative research [26, 27] and lacked empirical analysis to reveal the driving factors of the spatial distribution of national-level agricultural cultural heritage. In addition, the traditional methods of influencing factor research neglect the heterogeneous effects of factors on research units, making it difficult to fully analyze the degree of influence of each influencing factor on different research

units [28]. Therefore, this paper comprehensively utilizes various spatial analysis methods to deeply reveal the spatial distribution characteristics and patterns of China's national-level agricultural cultural heritage from 2013 to 2021. By combining multiple perspectives from governments, enterprises, community residents, and others, a comprehensive evaluation index system is constructed to uncover the internal causes and constraints of spatial distribution differences in agricultural cultural heritage, providing a scientific basis for policy formulation. This paper discusses the characteristics and evolution patterns of China's agricultural cultural heritage in spatial distribution. Based on the results of the influencing factor analysis, the author attempts to enumerate which regions in China have great potential for exploration, aiming to provide guidance for the excavation and protection of agricultural cultural heritage, enrich the theoretical system of agricultural cultural heritage research, promote the development of interdisciplinary research, and offer new perspectives and ideas for subsequent research.

## Data sources and processing and research methods

### Data sources and processing

The statistics of China's national agricultural cultural heritage come from the 138 agricultural cultural heritage sites in 6 batches from 2013 to 2021nnounced on the official website of the Ministry of Agriculture and Rural Affairs. Since an individual agricultural cultural site involves multiple districts and counties, when the latitude and longitude of each heritage location are obtained with the help of the Baidu API, they are processed separately to eventually obtain 144 latitudinal and longitudinal coordinates. Data on regional GDP, primary industry GDP, secondary industry GDP, general public budget expenditure, and per capita regional GDP are sourced from statistical yearbooks or bulletins of various provinces, cities, and counties.Population data are derived from spatial demographic statistics provided by Landscan, and some of the missing data are sourced from statistical yearbooks or bulletins of various provinces, cities, and counties.The data of the national key cultural relic protection units came from the national integrated online government affairs platform (the comprehensive administrative management platform of the State Administration of Cultural Heritage). The data of the national A-level scenic spots are from the official websites of the culture and tourism departments of the provinces, autonomous regions, and municipalities directly under the Central Government and the government affairs data resource websites. The precipitation and temperature data are from the ERA5-Land dataset released by the European Union and the European Center for Medium-range Weather Forecast. The original data are monthly average precipitation and temperature raster data. First, based on the original monthly precipitation and temperature raster, the raster calculation tool was used to calculate the mean values of the 12-month mean precipitation and temperature to obtain the mean precipitation and temperature raster data, and then, based on the mean precipitation and temperature rasters, the mean values of the raster values in each district and county were calculated. Through processing, the mean precipitation and temperature data at the county level were obtained. The elevation data were obtained from the Resources and Environmental Sciences and Data Center of the Chinese Academy of Sciences, and the river and road data were obtained from the National Geographic Information Resource Catalog Service System. The area statistical tool and the density calculation tool of ArcGIS 10.8 software were used to calculate the mean elevation of the county-level administrative districts. Value, river network density, and road density. Based on the above data, in the present study, 1,918 county-level administrative districts (excluding municipal districts and excluding agricultural cultural heritage sites) were used as the research unit, ArcGIS 10.8 software was used to classify the natural discontinuities in the data, and then geographic detectors were used for detection.

### Research methods

**Nearest neighbor index.** The principle of the nearest neighbor index (R) is to calculate the distance between each point feature and the nearest neighbor point in the calculation space. The R value is the ratio of the nearest neighbor distance to the theoretical nearest neighbor distance and is used to determine the spatial distribution of point features. aggregate type, random type, or uniform type [29]. The calculation formula is:

$$R = \frac{r}{r'} = 2\sqrt{D} \tag{1}$$

Where r is the actual nearest neighbor distance, r′ is the theoretical nearest neighbor distance, and D is the point density. When R = 1, the distribution of point features is random; when R>1, the point features are uniformly distributed; and when R<1, the point features have an agglomerated distribution.

**Centres for standard deviation ellipse analyses.** The center shift model is an important tool for analyzing the evolution of regional spatial patterns [30]. The moving direction and moving distance of the center of agricultural cultural heritage sites in each stage could be determined according to the changed trajectory of the center of agricultural cultural heritage sites. In 1926, Lefever proposed the standard deviation ellipse to represent the spatial distribution characteristics of geographic elements. The barycenter-standard deviation ellipse analysis used the barycenter, azimuth angle, major axis standard deviation, and minor axis standard deviation to analyze the spatial evolution characteristics of the research objects [27].

The formula for calculating the center of gravity is [31]:

$$x_j = \sum_{i=1}^{n}(G_{ij} \cdot x_i) / \sum_{i=1}^{n} G_{ij} \tag{2}$$

$$y_j = \sum_{i=1}^{n}(G_{ij} \cdot y_i) / \sum_{i=1}^{n} G_{ij} \tag{3}$$

where $(x_j, y_j)$ is the focus coordinate of the agricultural cultural heritage year, $(x_j, y_j)$ is the geometric center coordinate of the research unit, and $G_{ij}$ is the factor of the ith research unit and jth year.

The calculation formula for the offset distance of the center of gravity is:

$$D_{(j+1)-j} = C \times \sqrt{(y_{j+1} - y_j)^2 + (x_{j+1} - x_j)^2} \tag{4}$$

Where $(y_{j+1}, x_{j+1})$ and $(y_j, x_j)$ represent the coordinates of the center of gravity in different years, and C is a constant with a value of 111.11.

The orientation angle of the standard deviational ellipse:

$$\tan\theta = \left(\sum_{i=1}^{n} w_i^2 \tilde{x}_i^2 - \sum_{i=1}^{n} w_i^2 \tilde{y}_i^2\right)/2w_i^2\tilde{x}_i\tilde{y}_i + \sqrt{\left(\sum_{i=1}^{n} w_i^2 \tilde{x}_i^2 - \sum_{i=1}^{n} w_i^2 \tilde{y}_i^2\right)^2 + 4\sum_{i=1}^{n} w_i^2 \tilde{x}_i^2 \tilde{y}_i^2}/2w_i^2\tilde{x}_i\tilde{y}_i \tag{5}$$

The standard deviation of the major axis is:

$$\sigma_x = \sqrt{\sum_{i=1}^{n}(w_i\tilde{x}_i \cos\theta - w_i\tilde{y}_i \sin\theta)^2 / \sum_{i=1}^{n} w_i^2} \tag{6}$$

The standard deviation of the minor axis is:

$$\sigma_y = \sqrt{\sum_{i=1}^{n}(w_i\tilde{x}_i \sin\theta - w_i\tilde{y}_i \cos\theta)^2 / \sum_{i=1}^{n} w_i^2} \tag{7}$$

where $w_i$ represents the weight, $\theta$ represents the azimuth angle, and $(\tilde{x}_i, \tilde{y}_i)$ represents the coordinate deviation from the location of the research object $i$ to the center of the ellipse.

**Kernel density analysis.** Kernel density analysis. Kernel density analysis. Kernel density analysis can visually reflect the dispersion or aggregation characteristics of point features in geographic space [32], and its formula is:

$$f(x) = \frac{1}{n\text{h}}\sum_{i=1}^{n} \text{k}\left(\frac{x - x_i}{\text{h}}\right) \tag{8}$$

In the formula, N represents the number of agricultural cultural heritage sites in a region, h is the bandwidth, k is the kernel function, and $x - x_i$ is the distance from the estimated point to the kernel point.

**Spatial autocorrelation analysis.** Exploratory spatial analysis (ESDA) is used to test whether there is a significant association between a phenomenon and the attribute values of its neighboring units in space. Spatial autocorrelation is subdivided into global autocorrelation and local autocorrelation. Global autocorrelation is used to analyze the overall spatial correlation and agglomeration degree in a region [31], while local autocorrelation is a feature that describes the heterogeneity of elements [28]. The calculation formula for global autocorrelation is:

$$I = \frac{n}{s_o} \times \frac{\sum_{i=1}^{n} \sum_{j=1}^{n} w_{ij}(x_i - \bar{x})(x_j - \bar{x})}{\sum_{i=1}^{n}(x_i - \bar{x})^2} \tag{9}$$

Where n is the number of spatial units; $x_i$ and $x_j$ are the observed values of space units i and j, respectively; $w_{ij}$ is the spatial weight matrix, which is used to reveal the spatial connection between the elements; and $s_o$ is the aggregation of all spatial weights.

**Geo detectors.** Geographic detectors are tools for detecting and assessing spatial diversity. It includes four detectors: differentiation and factor detection, risk area detection, ecological detection and interaction detection [33]. In this study, with reference to the existing research results, after selecting the factors and indicators that affect the spatial distribution of China's important agricultural cultural heritage, the geographic detector method is used to detect and analyze the factors affecting the spatial distribution of agricultural cultural heritage. The formula is:

$$q = 1 - \left[\frac{\sum_{h=1}^{L} \sum_{i=1}^{N_h}(Y_{hi} - \bar{Y}_h)^2}{\sum_{i=1}^{N}(Y_i - \bar{Y})^2}\right] = 1 - \frac{\sum_{h=1}^{L} N_h \sigma_h^2}{N\sigma^2} = 1 - \frac{\text{SSW}}{\text{SST}} \tag{10}$$

$$\text{SSW} = \sum_{h=1}^{L} \sum_{i=1}^{N_h}(Y_{hi} - \bar{Y}_h)^2 = \sum_{h=1}^{L} N_h \sigma_h^2 \tag{11}$$

$$\text{SST} = \sum_{i=1}^{N}(Y_i - \bar{Y})^2 = N\sigma^2 \tag{12}$$

Where h is the number of layers of the impact factor, and $N_h$ and N are the number of units in

layer h and the whole area, respectively. $\sigma_h^2$ and $\sigma^2$ are the variances of layer h and the entire region, respectively. SSW and SST are the sum of the variances within the layer and the total variance in the whole area, respectively. The value range of q is [0,1].

## The evolution of the spatial pattern of national agricultural cultural heritage

### Type of spatial distribution

The average nearest neighbor analysis tool in ArcGIS 10.8 was used to calculate the nearest neighbor indices of the national-level agricultural cultural heritage sites in 2013, 2017 and 2021 to identify the spatial distribution type. The calculations revealed that the nearest neighbor indices for 2013, 2017 and 2021 were 1.07. The values of 0.71 and 0.75 indicate that the spatial distribution of national-level agricultural cultural heritage was discrete in 2013 (R>1) and showed a development trend of agglomeration in 2017 and later (R<1).

### Direction and range of spatial distribution

To further analyze the change in the spatial distribution of national agricultural cultural heritage over time, the present study used the center of standard deviation ellipse analysis model to calculate the dynamic change trajectories of the spatial distribution of agricultural cultural heritage in China in 2013, 2017 and 2021. The standard deviation ellipse and the migration trajectory of the center of gravity of agricultural cultural heritage during the sample period were calculated by ArcGIS 10.8 software (Fig 1) to describe the spatial distribution characteristics of national agricultural cultural heritage.

Based on Fig 1 and Table 1, the following specific analysis is carried out: (1) The spatial center of gravity shifts. From 2013 to 2021, the center of gravity of the spatial distribution of national-level agricultural cultural heritage moved between 111.996˚E-112.486˚E and 32.2106˚N-33.1546˚N; center of agricultural cultural heritage was located in Dengzhou city, Henan Province, in 2013, and it shifted northeastward by 79.23 km in 2017, with the center of gravity located in Wolong District, which will migrate southward by 104.99 km in 2021. The center of gravity is located in Zaoyang city, Hubei Province. The center of gravity is biased to the southeast in the azimuth, indicating that the spatial distribution of national-level agricultural cultural heritage shows a pattern of more in the southeast and less in the northwest. (2) Changes in the direction and range of spatial distribution. During the study period, the X-axis standard deviation of the national-level agricultural cultural heritage spatial distribution standard deviation ellipse was always greater than the Y-axis standard deviation, but the difference between the X and Y axes was relatively small, indicating that the overall spatial layout of national agricultural cultural heritage was northeast–southwest. The X-axis and Y-axis of the standard deviation ellipse showed a decreasing trend annually, and the area of the ellipse decreased from 4,197,300 km2 in 2013 to 3,469,900 km2 in 2017, indicating that there was a national-level agricultural cultural heritage space. The distribution range decreased to some extent, and the degree of dispersion gradually decreased, showing a trend toward centripetal aggregation. (3) Changes in the azimuth angle. From 2013 to 2021, the direction angle of the standard deviation ellipse decreased from 63.57˚ to 46.46˚, indicating a relatively small shift in the direction angle. This indicates that the direction of the spatial distribution of national-level agricultural cultural heritage is relatively stable, and the northeast–southwest direction is always the direction of national-level agricultural cultural heritage. The main direction of heritage space expansion.

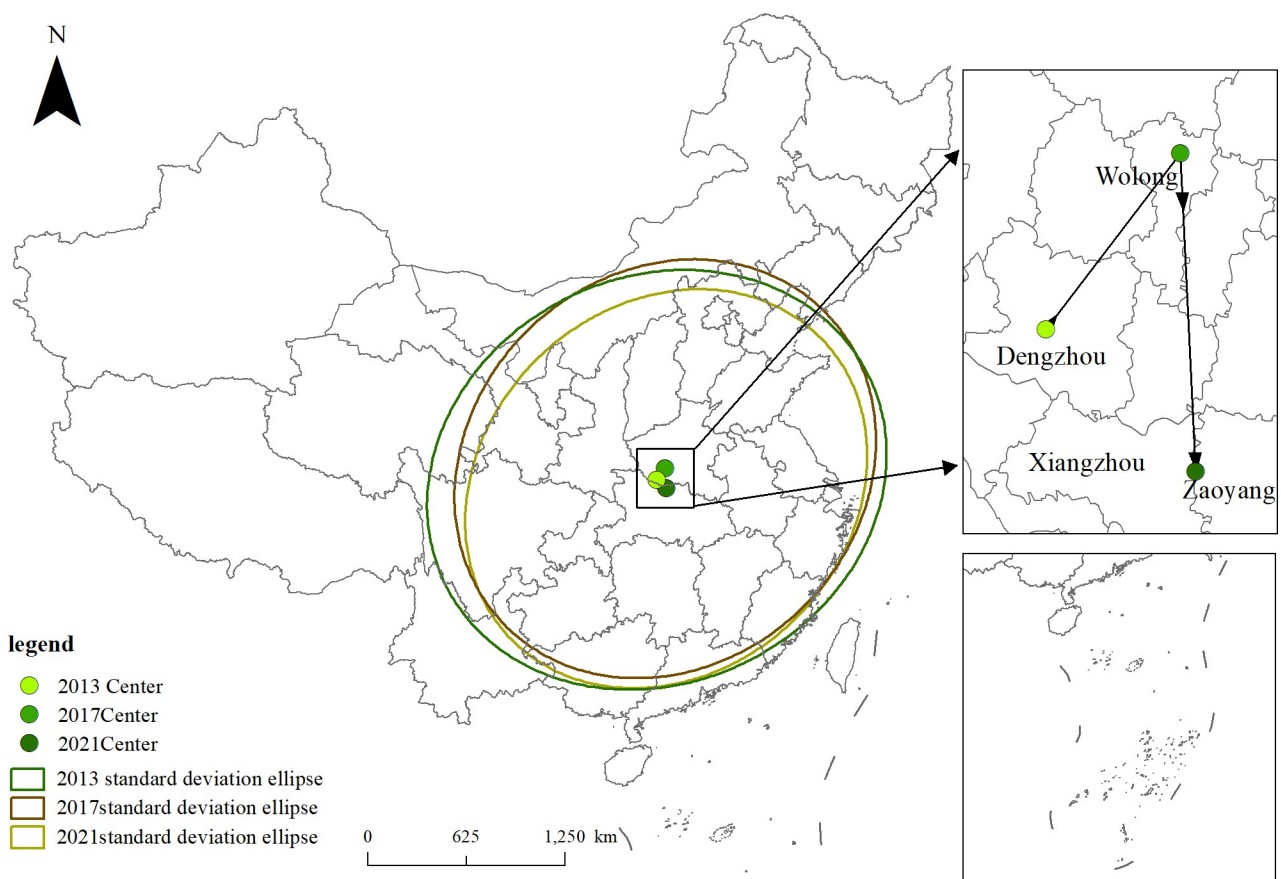

**Fig 1. Trajectory of the shift in the center of national agricultural cultural heritage and the distribution of standard deviation ellipses.**

## Overall distribution characteristics

According to the POI data of national agricultural cultural heritage sites, the spatial distribution of national agricultural cultural heritage sites during 2013–2021 was explored with the help of the kernel density analysis tool ArcGIS 10.8, and the spatial distribution density was analyzed. As shown in Fig 2, from 2013 to 2021, the distribution pattern of national-level agricultural cultural heritage gradually exhibited a "widely distributed, locally concentrated" distribution pattern, and the distribution of agricultural cultural heritage in the east and west increased significantly.

In 2013, the spatial distribution of national-level agricultural cultural heritage showed two obvious bands. The band with endpoints in southern Gansu and central Liaoning was located at the intersection zone of Northwest China and North China. Due to the many mountains and arid climate in this zone, agricultural cultural heritage sites are mainly composed of trees

**Table 1. Standard deviation ellipse of the spatial distribution of national agricultural cultural heritage.**

| Year | Centre of gravity coordinates | | X-axis standard deviation (km) | Y-axis standard deviation (km) | Area (million km²) | Direction Angle (°) |
|------|------|------|------|------|------|------|
| **2013** | 111.996205˚E | 32.661325˚N | 1246.99 | 1071.46 | 419.73 | 63.57 |
| **2017** | 112.511156˚E | 33.154585˚N | 1183.27 | 1031.61 | 383.47 | 46.50 |
| **2021** | 112.485624˚E | 32.210022˚N | 1141.25 | 967.87 | 346.99 | 46.46 |

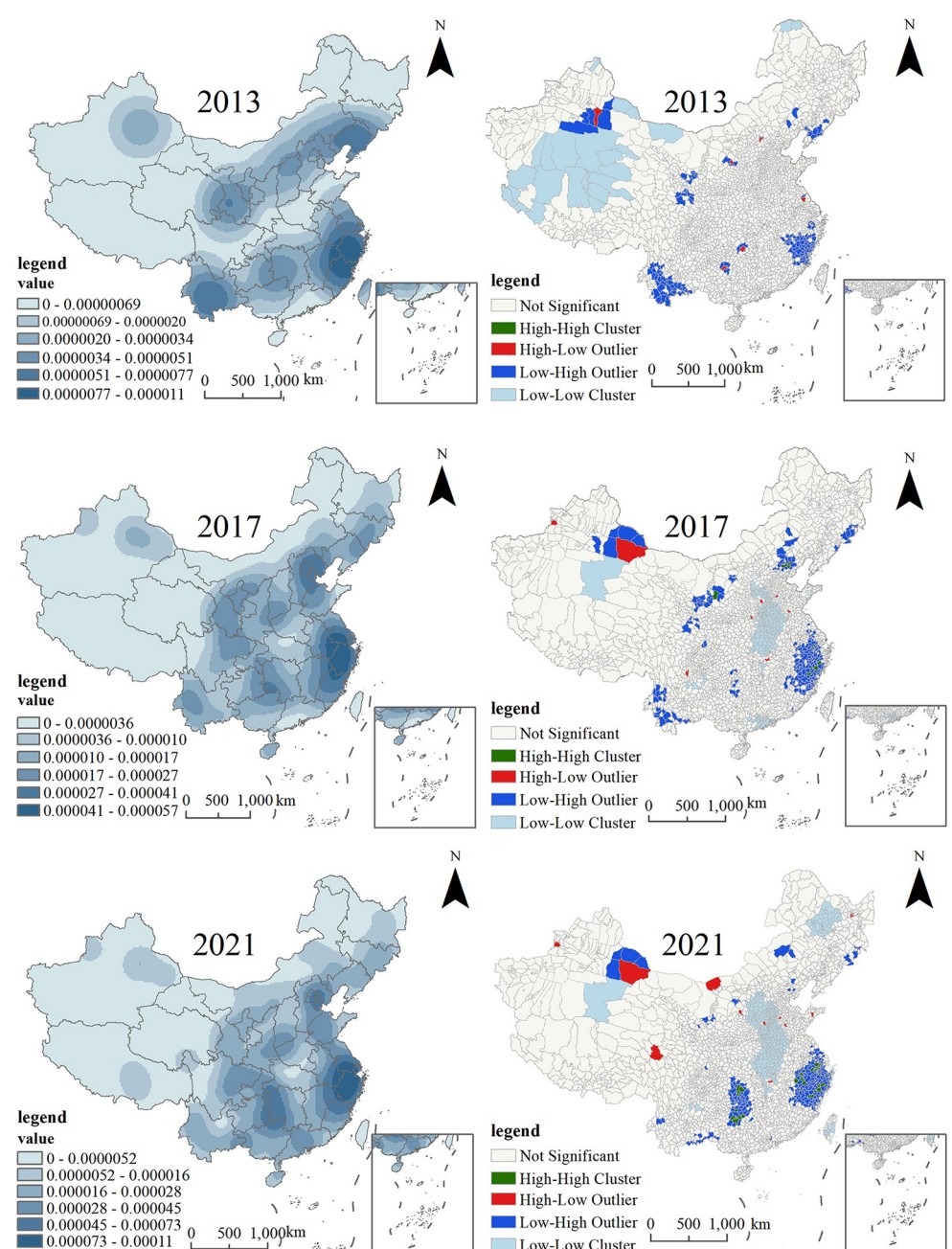

**Fig 2. Spatial distribution and clustering of the kernel density of national-level agricultural cultural heritage.**

and fruits. Next, a band is formed around the regions of Yunnan, Hunan, and Zhejiang. This band covers a high-density area and secondary density areas: the high-density core area is located at the edge of the interprovincial intersection of Fujian, Zhejiang and Jiangxi, and the types of agricultural cultural heritage are relatively diverse. The area is rich in farmland landscapes, composite systems, crop varieties, tea, and forest fruit systems. The reason is that this area is located in the southeastern coastal hilly area, a nationally important ecological functional area for biodiversity protection, and the coastal plain area, which has diverse terrains. Rich biodiversity and superior climatic conditions are conducive to the growth of different

agricultural varieties. The secondary density areas are located in Honghe Prefecture, Puer city, and the Bai Autonomous Prefecture in Yunnan Province. This area is located in the Hengduan Mountains, has a clear vertical climate, and is a multiethnic area. Rich culture.

Due to the further expansion of the distribution range of national-level agricultural cultural heritage, the spatial distribution density of agricultural cultural heritage underwent greater changes in 2017 to the extent that the original secondary density areas disappeared and new secondary density areas reproduced, i.e., the high-density core area. From the original edge area of the intersection of Fujian, Zhejiang and Jiangxi, it moved northward to the southern Jiangsu, eastern Anhui and Zhejiang areas. Because southern Jiangsu, eastern Anhui and Zhejiang are located in the Yangtze River Delta and are deeply influenced by the water culture south of the Yangtze River, they have a great impact on various agroecosystems, such as rice culture, tea culture, fishery culture, etc., and at the same time, they are located in the Yangtze River Basin. The unique geographic resources provide a unique geographic environment for agricultural development, which, together with the strong economic strength of these areas, can provide sufficient financial support for agricultural development. Therefore, due to the comprehensive effect of many factors, the amount of agricultural cultural heritage in this area continues to increase. It has become a high-density core area of various types, mainly the composite system and the agricultural cultural heritage of forest and fruit. In addition, the degree of agglomeration of the Beijing-Tianjin-Hebei region has become increasingly obvious, becoming a secondary density area. As an economically developed urban agglomeration in China, this area has strong economic strength and is deeply influenced by agricultural culture and dry land farming culture, mainly.

Statistics on the types of national-level agricultural cultural heritage showed that in 2021, China's agricultural cultural heritage was dominated by forest and fruits, with 49 items accounting for 36% of the total number of items, indicating that forest and fruit agricultural products play an important role in China's agricultural production process. China has an advantage. China has a vast territory with complex and diverse terrain, straddling five temperature zones from north to south, and has various climate types with great differences between regions in all aspects, which can provide suitable growing conditions for different forest and fruit agricultural products. The high-density area is still located in the Yangtze River Delta, and the types of agricultural cultural heritage are still based on composite systems and forest and fruit heritage; among them, there are 6 composite heritages and 7 forest and fruit heritages, forming the "Composite System of Tea and Fruit in Biluochun, Wuzhong, Jiangsu." system, Jiangsu Gaoyou Lake wetland agricultural system, Jiangsu Xinghua duotian traditional agricultural system, Zhejiang Jinyun bamboo shoot-shelduck symbiosis system, Zhejiang Qingtian rice-fish symbiosis system, Anhui Taihu mountainous compound agriculture system, Jiangsu Wujiang sericulture culture system, and Zhejiang Shaoxing Society. There are 27 national-level agricultural cultural heritage items, including those of the Jishan Torreya Group. At this stage, the number of agricultural cultural heritage sites in the Beijing-Tianjin-Hebei region was only one item higher than that in 2017, and the growth rate was relatively slow compared to that in other regions, indicating that the excavation of agricultural cultural heritage sites was not sufficient. Agglomeration of the border area at the intersection of Guizhou, Hunan and Yunnan has occurred. This area is located in the Wuling Mountain area, the old revolutionary base areas, and the gathering places of multiple ethnic groups. Ethnic minorities have created and developed unique ethnic cultures through long-term living practices, and this area has a unique ecology. The environment and rich lifestyle and agricultural activities included purple magpie terraced fields in Xinhua, Hunan; the Huayuanzi-Lagong rice composite planting system in Hunan; the rice-fish-duck composite system in Congjiang Dong Township, Guizhou; and the traditional planting and management system of Jinping fir in

Guizhou. These activities involve the crystallization of the wisdom of ethnic minorities in the process of agricultural production.

Overall, the spatial distribution of China's national agricultural cultural heritage in China is influenced not only by the natural environment and the level of economic development but also by ethnic and regional cultures. The spatial distribution of agricultural cultural heritage is unbalanced, and there are large differences in regional spatial distribution, which has a strong tendency. Nationality and regionality. Among the 138 existing agricultural cultural heritage items, there are 46 items in ethnic minority areas, accounting for 33% of the national total. The types of items are rich. Among them, there are 14 agricultural cultural heritage items in the southwestern region. There are more ethnic minorities in the southwestern region, and a splendid ethnic culture has been formed through the blending of multiple ethnic groups in the southwestern region. Coupled with the complicated topography and unique ecological environment, the number of ethnic minorities in the southwestern region is greater than that in the northwestern and northeastern regions. Second, China's national agricultural cultural heritage is widely distributed in coastal areas, with more coastal areas and fewer inland areas. There are 56 agricultural cultural heritages in coastal areas, with Zhejiang having the most prominent number of items, reaching 14. Traffic accessibility in coastal areas is strong. They have frequent exchanges with the outside world, strong cultural inclusiveness, and a high awareness of agricultural cultural protection; occupy a dominant position in physical geography, social culture, and economy; and can provide a strong force for the protection and development of agricultural cultural heritage. There are more agricultural cultural heritages in the Changjiang and Yellow River Basins, and the number of agricultural heritages outside these basins is relatively small. The reason is that these two river basins are the origin of China's farming culture, which is characterized by a profound farming culture, rich agricultural resources, and a long history of agricultural development, and these basins have accumulated a rich agricultural cultural heritage. Production experience. In addition, the study revealed that, as major agricultural provinces in China, Henan, Hubei, and Heilongjiang have rich agricultural resources and a long agricultural history, but the number of agricultural cultural heritage items is relatively small, which shows that the emphasis on the excavation and protection of agricultural cultural heritage in some provinces is insufficient attention and resource investment, and the full potential of agricultural cultural heritage has not been tapped. Shanghai and Qinghai are areas lacking agricultural cultural heritage. The reason is that, as China's economic center and international metropolis, Shanghai's agricultural development is mainly based on urban agriculture and modern agriculture. With the acceleration of urbanization and the introduction of modern agricultural technology, the promotion and restructuring of the agricultural industry have caused some traditional agricultural technologies and production methods to be phased out, which has a great impact on the production models of traditional agriculture. Qinghai Province is located on the Qinghai–Tibet Plateau of China, and its unique geographic environment and climatic conditions make it an alpine province. This area is characterized by dryness, abundant sunshine and a lack of heat, and it faces enormous challenges in terms of economic development, population distribution, and transportation conditions. These factors greatly constrain the development of agriculture.

## Spatial association features

To explore the spatial distribution characteristics of national agricultural cultural heritage sites in more detail, districts and counties were used as research units to analyze the agglomeration degree and spatial heterogeneity characteristics of national agricultural cultural heritage sites. In this study, ArcGIS 10.8 software was used to analyze Moran's I index (Table 2). The analysis

**Table 2. Global autocorrelation coefficients of spatial patterns of national agricultural cultural heritage sites.**

| year | Moran I | Z value | P value |
|---|---|---|---|
| 2013 | 0.0012 | 0.4475 | 0.654 |
| 2017 | 0.0061 | 1.8253 | 0.068 |
| 2021 | 0.0152 | 4.4198 | 0.000 |

results showed that all Moran's I indices were positive and passed the significance level test of 10% or 1% (except for 2013). This shows that after 2017, agricultural cultural heritage had a positive spatial correlation and gradually increased. The spatial distribution of agricultural cultural heritage of each research unit is not random but can influence each other within a certain range to form a spatial agglomeration effect.

Second, to further clearly show the correlation of the spatial distribution of agricultural cultural heritage at the national level, the present study used the clustering and outlier analysis tools in ArcGIS 10.8 software to perform local autocorrelation analysis on the distribution of agricultural cultural heritage in each research unit, and four types of local spatial agglomeration were constructed (Fig 2), namely, HH, HL, LL and LH. Among them, the HH (LL) type area was a high-value agglomeration area (low-value agglomeration area), indicating a positive spatial correlation; the HL (LH) type area was an anomalous area with high values containing low values (a low value contained an anomalous area with high values), indicating a negative spatial correlation. The analysis results show that the spatial agglomeration of national-level agricultural cultural heritage from 2013 to 2021 is dominated by LL and LH type areas, while the HH and HL type areas are distributed in relatively small numbers.

The HH type area is a state of mutual agglomeration of high-value areas, and its distribution is basically consistent with the distribution of high-density areas of agricultural cultural heritage. The HH type areas began to appear in 2017, with a small number of distributions involving 10 county-level cities, including Yanchi County, Xinglong County, Qianxi County, Qingyuan County, and Qingtian County. In 2021, the HH type areas were clustered in the Yangtze River Delta and the edge areas at the intersection of Guizhou, Hunan and Yunnan, covering 31 county-level cities, including Congjiang County, Songxi County, and Qingyuan County, indicating that the spatial agglomeration of national-level agricultural cultural heritage gradually increased. Among them, 21 HH type areas were identified in the delta area of Changjiang County, accounting for 68% of the total number of HH type areas, which further verified that the Yangtze River Delta region is the most important gathering place of agricultural cultural heritage in China.

The HL type area showed high values for itself and low values for the surrounding areas, indicating a significant negative spatial correlation. From 2013 to 2021, this type of area was distributed in a scatter-like manner in space, with a small number of distributions and a relatively stable spatial layout. In 2013, the HL type area was mainly distributed near the LH type area, and after 2017, it was mainly distributed around the LL type area.

Before 2017, the LH type areas were mainly distributed in the Yangtze River Delta region, including Gutian County, Lanxi city, Pujiang County, and Wuyi County, and in Yunnan, including Yongping County, Lvchun County, and Jinggu Tai and Yi Autonomous County. After 2021, its spatial distribution showed a trend of agglomeration, which was mainly concentrated in the county cities in the Yangtze River Delta and the edge areas at the intersection of Guizhou, Hunan and Yunnan and was distributed in sheets. In the Yangtze River Delta, Anji County, Wujiang District, Wuzhong District, Huangyan District, Eight Districts and Counties, including Jinyun County, Songxi County, Kaihua County, and Tongxiang City, changed from

the LH type area to the HH type area, indicating that the agglomeration of national-level agricultural cultural heritage in these Yangtze River Delta regions significantly increased.

The LL-type areas were mostly distributed in areas with relatively backward economic development, and the spatial distribution pattern changed greatly during the study period. In 2013, it was mainly distributed in the intersection area of Xinjiang, Tibet, and Qinghai provinces, including Qiemo County, Gaize County, Shuanghu County and Anduo County. This area does not have advantages in terms of nature, economy, social culture, or agricultural development; in 2017, these areas were mainly distributed in central areas such as Zaoyang city, Sui County, Fengqiu County, Xihua County, and Qi County, as well as southern China such as Shunde District, Boluo County and Deqing County. The central area is a low-lying area with a spatial distribution of agricultural cultural heritage. The number of heritage areas was small; in 2021, LL-type areas will continue to expand to the county cities in the central region, LL-type areas will begin to appear in Northeast China, and LL-type areas will decrease significantly in South China. This is because in recent years, Guangdong has strengthened its inheritance, protection, and promotion of farming culture. Gradually, the protection and utilization of important agricultural cultural heritage sites have gradually increased. In 2014, only the Fenghuang dancong tea culture system in Chaoan, Guangdong, was selected as an important agricultural cultural heritage site in China, and by 2021, six agricultural cultural heritages were selected.

## Driving factors of the evolution of the spatial pattern of national-level agricultural cultural heritage

### Selection of influencing factors

The spatial distribution of national agricultural cultural heritage is influenced by numerous factors. Drawing upon relevant research on cultural heritage, this study analyses the influencing factors of the spatial distribution of national agricultural cultural heritage at four major levels namely, the government, community residents, enterprises, and others, encompassing 13 specific factors (Table 3). The mechanism of their influence is illustrated in Fig 3.

The difference in economic development among regions is an important factor affecting the unbalanced spatial distribution of national agricultural cultural heritage [26]. Areas with a higher level of economic development have stronger economic strength and technological

**Table 3. Factors influencing the spatial distribution of national agricultural cultural heritage.**

| Index layer | Factor | No. | Evaluation factor | Unit |
|---|---|---|---|---|
| **Government level** | Regional Economic Development Level | X1 | Gross Regional Product | ten thousand yuan |
| | Policy Guarantee | X2 | General Public Budget Expenditure | ten thousand yuan |
| | Traffic Accessibility | X3 | Highway density | km/km2 |
| | Cultural Environment | X4 | Number of national key cultural relics protection units | a/an |
| **Community Level** | Per capita economic development level | X5 | GDP per capita | million/person |
| | Population | X6 | Total population | per person |
| **Enterprise Level** | Primary Industry Economic Development Level | X7 | Value Added of Primary Industry | % |
| | Tertiary Industry | X8 | Value-added ratio of tertiary industry | % |
| | Tourism Resource Endowment | X9 | Number of A-class scenic spots in China | a/an |
| **Others** | Temperature | X10 | Average temperature | ˚C |
| | Precipitation | X11 | Average precipitation | m |
| | Topography | X12 | Average elevation value | m |
| | Rivers | X13 | Density of river network | km/km2 |

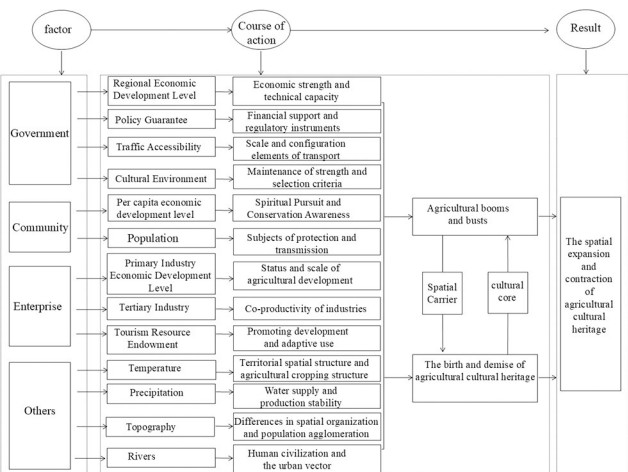

**Fig 3. Mechanisms influencing spatial patterns of agricultural cultural heritage.**

capabilities and thus more resources for the protection and development of agricultural cultural heritage. The protection, inheritance and active utilization of agricultural cultural heritage are inseparable from the support of relevant policies [34], and fiscal expenditure is an important regulating means for the sustainable development of agriculture and agricultural cultural heritage [35]. The scale of transportation and the allocation of elements needed for cultural heritage improvement in transportation accessibility affect the inheritance and protection of culture [36]. Culture is an important force in maintaining the agricultural landscape of cultural heritage sites [37] and is one of the core selection criteria for national agricultural cultural heritage [10]. National cultural relic protection units are an important part of cultural heritage [38] and reflect the cultural accumulation and historical heritage of a region. The government plays an important role in regional economic development, financial support, transportation accessibility, and cultural relic protection. Therefore, the present study selects GDP, general public budget expenditures, road density, and the number of key cultural relic protection units nationwide to characterize government-level contributions. The level of regional economic development, policy guarantees, transportation accessibility, and the cultural environment.

The generation, development, and dissemination of agricultural cultural heritage are closely related to human activities [39]. The development of agriculture depends on human production activities, and people are the most active and core factor in the protection and development of agricultural cultural heritage [26]. With the gradual improvement of people's living standards, people's spiritual needs are also increasing. This spiritual need is reflected not only in the pursuit of material matters but also in the pursuit of cultural accomplishment. Therefore, under this trend, people's interest in agriculture and awareness of the concern and protection of cultural heritage will increase. Therefore, the per capita GDP and total population were chosen to represent the per capita economic development level and demographic status at the community resident level, respectively.

The gross product value of the primary industry reflects the degree and scale of agricultural development and is an important indicator for measuring the status of agricultural economic development in a country or region. Agriculture and other industries are jointly produced. Agriculture is the carrier of resources and has direct or indirect links with other industries [40]. The development of tourism resources can promote the active utilization of agricultural cultural heritage, and the development of tourism is helpful for its maintenance and display.

Liu et al. [11] showed that there is a significant positive correlation between agricultural cultural heritage and tourism development. The operational efficiency, technological innovation, and market strategy of enterprises also play important roles in the added value of primary industry; enterprises show high activity and innovation in tertiary industry; and tourism enterprises play an important role in the development of scenic resources and in management and operation. Therefore, the proportion of the added value of the primary industry, the proportion of the added value of the tertiary industry and the number of A-grade scenic spots nationwide were selected to characterize the economic development level of the primary industry, the economic development level of the tertiary industry and the endowment of tourism resources at the enterprise level.

Temperature and precipitation are resource endowments for agricultural production. Suitable temperature and precipitation conditions are conducive to human settlement and agricultural production [41]. Temperature differences shape the regional spatial structure and profoundly affect the agricultural planting structure. Water is an important source of agricultural production, and its abundance is directly related to the stability of agricultural production. Different surface morphologies form different natural textures and spatial organizations [36], which is one of the selection criteria for population settlement and affects the formation and development of agricultural and cultural heritage. Natural waters such as rivers have comprehensive functions, such as agricultural irrigation, fishery production, and domestic water supply, and are closely connected with human settlements [41]. Rivers are the birthplace of human civilization and the carriers of cities [42]. Natural conditions play a decisive role in the location selection of villages. The formation of villages in turn gives rise to a series of production activities, shaping the unique agricultural cultural heritage of each region.

## Results analysis of the influencing factors

**Factor exploration analysis.** Based on the data processing results of the spatial distribution of national agricultural cultural heritage, a geographic detector was used to determine the explanatory power (q value and p value) of each influencing factor on the spatial distribution of national agricultural cultural heritage. The detection results (Table 4) show that, except for river and population status, which passed the 5% significance level test, the remaining factors all passed the 1% significance level test, and their explanatory power was in the order of tourism resource status (0.060), regional economic development level (0.042), policy security

Table 4. Detection of influencing factors of spatial distribution of national agricultural cultural heritage.

| Index layer | Factor | No. | q value | P value |
|---|---|---|---|---|
| Government level | Regional Economic Development Level | X1 | 0.042 | 0.000 |
| | Policy Guarantee | X2 | 0.039 | 0.000 |
| | Traffic Accessibility | X3 | 0.013 | 0.000 |
| | Cultural Environment | X4 | 0.018 | 0.000 |
| Community Level | Per capita economic development level | X5 | 0.018 | 0.000 |
| | Population | X6 | 0.026 | 0.016 |
| Enterprise Level | Primary Industry Economic Development Level | X7 | 0.025 | 0.000 |
| | Tertiary Industry | X8 | 0.012 | 0.007 |
| | Tourism Resource Endowment | X9 | 0.060 | 0.000 |
| Others | Temperature | X10 | 0.012 | 0.003 |
| | Precipitation | X11 | 0.019 | 0.000 |
| | Topography | X12 | 0.015 | 0.000 |
| | Rivers | X13 | 0.010 | 0.025 |

(0.039), demographic status (0.026), level of economic development of the primary industry (0.025), precipitation (0.019), cultural environment (0.018), per capita level of economic development (0.018), and topography (0.015). The indicators included transportation accessibility (0.013), temperature (0.012), the level of economic development of tertiary industry (0.012), and rivers (0.010).

Among many influencing factors, enterprise-level factors (0.032) have a relatively significant effect on the spatial differentiation of national-level agricultural cultural heritage, indicating that enterprises are the leading force in the development of agricultural cultural heritage. The explanatory power of tourism resources was 0.06, indicating that the status of tourism resources is the main driving force of the spatial pattern of national agricultural cultural heritage. There is a definite correlation between the richness of tourism resources and the protection and inheritance of agricultural cultural heritage. Rich tourism resources often mean that an area has unique natural landscapes, profound cultural heritage or historical relics, and relatively complete infrastructure. To maintain the stability of tourism development, relevant local ministries will closely track and actively respond to people's travel needs and tap potential local tourism resources. As a new form of tourism, the development and utilization of agricultural cultural heritage tourism will receive attention from various departments.

The explanatory power of the spatial distribution of agricultural cultural heritage at the government level was 0.028, indicating that the government is the driving force for the development of agricultural cultural heritage. In the process of developing and protecting agricultural cultural heritage, the government plays a role in policy guidance and support, capital investment and resource integration, heritage selection and supervision and management, and interest coordination and sharing, providing clear guidance and planning for the development of agricultural cultural heritage. The explanatory power of the level of economic development was 0.042, second only to the status of tourism resources. These results suggest that the development and protection of agricultural cultural heritage depend on the local economic development level. The possible reason is that areas with higher levels of economic development have stronger protection awareness, and economic strength and scientific and technological strength have laid a solid foundation for the formation and development of agricultural cultural heritage. Second, policy guarantees are also one of the keys to the development of agricultural cultural heritage, with an explanatory power of 0.039. Fiscal expenditure can provide financial support, project funding, and infrastructure construction for the protection and inheritance of agricultural cultural heritage and promote the development of agricultural cultural heritage-related work. Transportation, which helps people understand and experience agricultural culture and enhances its vitality, is an important channel for disseminating agricultural culture. Regional culture, on the other hand, has a nourishing effect on the generation and development of agricultural cultural heritage and thus is a fertile soil for the growth of agricultural cultural heritage and the development of regional culture. Therefore, the role of transportation and the cultural environment in the formation and development of agricultural cultural heritage cannot be ignored.

The explanatory power of the spatial distribution of agricultural cultural heritage at the community resident level was 0.022, indicating that community residents are an important force in the development of agricultural cultural heritage. The prosperity and development of agriculture is the space carrier of agricultural cultural heritage, and the labor force is an indispensable part of agricultural development. Therefore, the breeding, protection and prosperity of agricultural cultural heritage are deeply rooted in agricultural production activities passed down by human beings from generation to generation.

The explanatory power of physical and geographic factors for the spatial distribution of agricultural cultural heritage was 0.014. Differences in natural and geographic environments

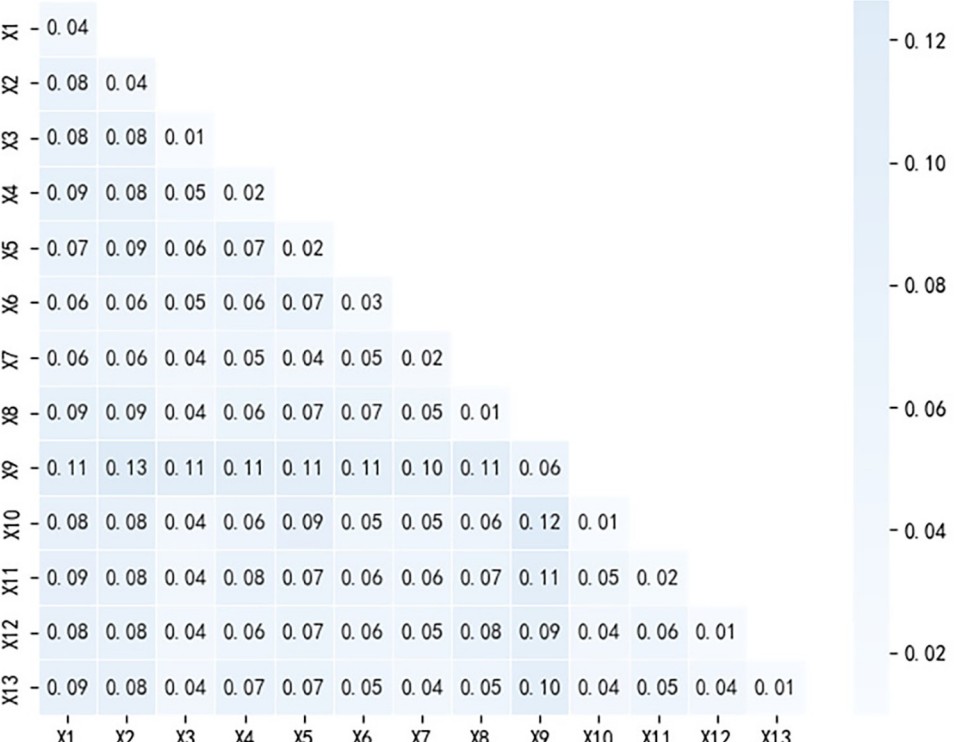

**Fig 4. Interaction detection of factors influencing the spatial distribution of national agricultural cultural heritage.**

have created different types of agricultural cultural heritage. Among the influencing factors of physical geography, precipitation has the greatest impact on the spatial distribution of agricultural cultural heritage, followed by topography and temperature. According to the above analysis, national-level agricultural cultural heritage is mainly distributed in the eastern region. Most of these plains and hills, which have low altitudes, gentle terrain, suitable climates, and fertile soil and water, are conducive to the development of agriculture and the protection of heritage sites.

**Factor interaction detection analysis.** Pairwise interaction detection was performed on the 13 factors that affect the spatial distribution of agricultural cultural heritage at the national level (Fig 4). The results showed that the explanatory power of the interaction between any two factors was greater than that of a single factor, and the influencing factors interacted with each other. The spatial distribution of agricultural cultural heritage at the national level is not affected by a single factor but by the interaction of various factors, except for the interactions of X1 and X2, X6, X7, X2 and X6, and X7 and X5. The intersection of X7 showed dual-factor enhancement, and the remaining factors all exhibited a nonlinear enhancement relationship. In the interaction between any two factors, the interaction between the tourism resource status and the remaining 12 factors had a more significant impact on the spatial distribution of agricultural cultural heritage, and most of the factors showed that the interaction between the two factors had a significant impact on the spatial distribution of agricultural cultural heritage. The $q$ value is greater than 0.1, followed by the interaction between the level of regional economic development and policy security and the remaining factors. The explanatory power of the single factor of tourism resource status is relatively strong, and the interaction with other factors is more significant, indicating that the influence of tourism resources not only exists

independently but also interact and reinforce each other with other factors. This may be due to the comprehensive nature of tourism resources, so the formation and development of tourism resources are comprehensively affected by natural environmental factors such as climate, topography, and rivers, as well as human factors such as economic development level, population status, and cultural environment. Therefore, there was a significant interaction effect with other factors. The level of regional economic development was the secondary driving force for the spatial pattern of national agricultural cultural heritage, and interactions with other factors had a greater impact. Areas with higher levels of economic development could attract the inflow of factors such as population and capital, thus promoting the construction of relevant infrastructure. To a certain extent, the development of agriculture and agricultural cultural heritage should be promoted. Therefore, in the development, protection and development of national agricultural cultural heritage, targeted and synergetic strategies should be adopted based on the interaction intensity of the factors.

## Discussions and conclusions

### Discussions

Most scholars' studies have analyzed the spatial distribution characteristics of national agricultural cultural heritage based on a static time cross-section. For example, Liu Haitao and his colleagues analyzed the spatial distribution characteristics of the 59 globally important agricultural heritage systems (GIAHS) in 2020 using the Gini coefficient method, from the perspectives of the Eastern and Western Hemispheres, the Northern and Southern Hemispheres, as well as continents and countries [43]. Han Zongxin conducted a spatial distribution characteristic analysis of China's agricultural cultural heritage in 2015 using kernel density analysis and hot spot exploration [24]. This tends to ignore the dynamic evolution of the process as well as the long-term trend of change and fail to fully reveal its complexity over time. This article takes the period from 2013 to 2021 as the research perspective and dynamically reveals the evolution of the spatial pattern of national agricultural cultural heritage. In terms of influencing factors, based on empirical analyses, this paper selects county-level administrative districts as the research unit to explore in depth the driving factors of the spatial pattern of national agricultural cultural heritage with a micro vision and the research methodology employed geographical detectors for analysis, distinguishing it from previous qualitative studies.

Due to the lack of long time series impact data, this paper explores only the factors influencing the spatial distribution of national agricultural cultural heritage sites in 2021.As a result, it was not possible to explore the drivers of national agricultural cultural heritage in different years, and the shortcomings in this area should be addressed in future research.

### Conclusion and future prospects

In this paper, spatial analysis methods such as the nearest-neighbor index, Centres for standard deviation ellipse analyses, kernel density, spatial autocorrelation and geographic probes are used to explore the spatial distribution characteristics and driving factors of national agricultural cultural heritage. The specific conclusions are as follows:

1. After 2017, the nearest neighbor index of national-level agricultural cultural heritage was less than 1, indicating that during 2013–2021, national-level agricultural cultural heritage gradually agglomerated in spatial distribution. The spatial distribution center was "northeast first, then southeast", with the center of agricultural cultural heritage biased to the southeast in the central azimuth, indicating that the national-level agricultural cultural

heritage exhibited a distribution pattern of more southeast than northwest. The area of the standard deviation ellipse gradually decreased, the centripetal agglomeration trend increased, the azimuth offset was relatively small, and the northeast–southwest direction has always been the main direction of the expansion of national-level agricultural cultural heritage space.

2. The spatial pattern of national-level agricultural cultural heritage changed greatly from 2013 to 2021, showing a gradual distribution pattern of "widely distributed and locally concentrated"; the high-density core area of national-level agricultural cultural heritage moved northward from the original edge area at the intersection of Fujian, Zhejiang and Jiangxi. In the Yangtze River Delta, there are various types of agricultural cultural heritage, with the agricultural cultural heritage of the composite system and forest and fruit being the main heritage. The spatial distribution of national-level agricultural cultural heritage is unbalanced, and the regional spatial distribution varies greatly, showing strong nationality and regional characteristics. Compared with other regions such as Northeast China, there are more national-level agricultural cultural heritages of ethnic minorities, there are extensive branches in coastal areas and fewer inland areas, and there are more people in the Changjiang and Yellow River basins and fewer people outside the basins. In addition, the study results show that there is a certain lag in the excavation and protection of agricultural cultural heritage in Henan, Hubei, and Heilongjiang. These findings were consistent with the conclusions of previous studies.

3. During 2013–2021, the spatial correlation and heterogeneity of national-level agricultural cultural heritage gradually became significant; the spatial distribution types were mainly LL and LH types; during the study period, the distribution of LL- and HH-type areas continued to increase, and the positive spatial correlation gradually increased. Enhanced. HH-type areas began to appear in 2017 and were mainly distributed in county-level cities in the Yangtze River Delta, involving a relatively small number of county-level cities, including 10 county-level cities, and by 2021, involving 31 county-level cities. HL-type areas were scattered spatially. The distribution was dot-like, and the distribution number was small. The LH-type areas showed a trend of agglomeration and were mainly concentrated in the county-level cities at the edge of the Yangtze River Delta and at the intersection of Guizhou, Hunan and Yunnan. The LL-type areas were mostly distributed in areas with relatively backward economic development. The pattern changed significantly.

4. According to the results of geographic detectors, the spatial pattern of national-level agricultural cultural heritage is affected by factors such as temperature, precipitation, level of economic development, population status, transportation accessibility, cultural environment, and the status of tourism resources. The explanatory power of developmental level was relatively significant. These factors interact closely, and the explanatory power of the interaction is greater than that of a single factor. Therefore, the spatial distribution pattern of agricultural cultural heritage at the national level is affected by the interaction of various factors.

Countermeasures and suggestions:

1. The in-depth discovery and comprehensive protection of agricultural cultural heritage in ethnic minority areas should be promoted. The results show that national-level agricultural cultural heritage projects in ethnic minority areas account for 33% of the country's total, indicating that agricultural cultural heritage in ethnic minority areas has great potential and potential for development. Qinghai, Tibet and Heilongjiang, which are important ethnic

minority regions in China, have unique ecological environments and rich lifestyles as well as agricultural activities and are rich in potential resources for agricultural cultural heritage. Examples include the Tibetan Plateau sheep breeding system and the Tibetan Plateau yak breeding system in Qinghai, the agro-forestry-pastoral composite system in the southeastern Tibetan canyon area, the traditional forest management system in the Linzhi area, and the ancient salt well salt field system in Mangkang in Tibet. However, as of 2022, there are no agricultural cultural heritage projects in Qinghai Province and only two in Tibet and Heilongjiang. Therefore, in future assessments of agricultural cultural heritage, it is necessary to provide appropriate policy preferences to ethnic minority regions, help with the declaration and selection of agricultural cultural heritage in these regions, and promote the coordinated development of agricultural cultural heritage in the country.

2. Strengthening interprovincial exchanges on the development and protection of agricultural cultural heritage, establishing interprovincial cooperation demonstration zones, and exploring interregional modes of joint protection and development. During the study period, national agricultural cultural heritage sites showed a clear tendency to cluster in the interprovincial intersection areas of Fujian, Zhejiang, Jiangxi, Beijing, Tianjin, Hebei, Guizhou, Hunan, Yunnan, Anhui, Zhejiang and Suzhou. This clustering tendency not only reflects the geographical characteristics of agricultural cultural heritage but also indicates that the fringe area of the interprovincial intersection may become a golden zone for the clustering of agricultural cultural heritage. Through the establishment of interprovincial cooperation demonstration zones, geographical constraints will be broken down, resources from different regions will be integrated, resource waste will decrease, and the efficiency of heritage conservation and development will improve. In addition, experts, scholars and practitioners from various provinces can be invited to discuss protection strategies, development models and innovative cases of cross-regional agricultural cultural heritage by organizing seminars and forums on the protection and development of agricultural cultural heritage.

3. The tourism value of agricultural cultural heritage should be further investigated, and the integration and development of national agricultural cultural heritage and tourism should be promoted. The study revealed that the richness of tourism resources has a significant impact on the spatial distribution of national agricultural cultural heritage, which indicates that there is a close interactive relationship between tourism and the development of agricultural cultural heritage. Agricultural cultural heritage, as a treasure of farming civilization, carries rich historical information and cultural connotations, and its diversity is reflected not only in the unique ways of agricultural practice but also in folk customs, traditional crafts and ecological wisdom. Therefore, in focusing on the excavation and protection of agricultural cultural heritage, we should actively explore mass tourism and study the tourism of agricultural cultural heritage, develop suitable tourism routes, create cultural and creative products with special characteristics, increase the social attention given to agricultural cultural heritage, and enhance people's awareness of protection to realize a win–win strategy for the protection and economic development of national agricultural cultural heritage.

## Supporting information

**S1 File. Research data.**
(XLSX)

## Author Contributions

**Conceptualization:** Yuyu Liao, Jun Liu.

**Data curation:** Yuyu Liao, Jun Liu.

**Formal analysis:** Yuyu Liao, Jun Liu.

**Investigation:** Yuyu Liao.

**Methodology:** Yuyu Liao.

**Supervision:** Lifei Xu, Jun Liu.

**Validation:** Yuyu Liao, Jun Liu.

**Visualization:** Yuyu Liao.

**Writing – original draft:** Yuyu Liao.

**Writing – review & editing:** Lifei Xu, Jun Liu.

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
