## [Decision Letter · Decision Letter 0]

13 Sep 2024

PONE-D-24-22604Study on the Evolution of the Spatial Patterns and Driving Factors of National Agricultural Cultural Heritage in ChinaPLOS ONE

Dear Dr. LIU,

Thank you for submitting your manuscript to PLOS ONE. After careful consideration, we feel that it has merit but does not fully meet PLOS ONE’s publication criteria as it currently stands. Therefore, we invite you to submit a revised version of the manuscript that addresses the points raised during the review process.

We look forward to receiving your revised manuscript.

Kind regards,

Bijay Halder

Academic Editor

PLOS ONE

3. In the online submission form, you indicated that your data is available only on request from a third party. Please note that your Data Availability Statement is currently missing the name of the third party contact or institution / contact details for the third party, such as an email address or a link to where data requests can be made. Please update your statement with the missing information.

4. We note that Figures 1 and 2 in your submission contain [map/satellite] images which may be copyrighted. All PLOS content is published under the Creative Commons Attribution License (CC BY 4.0), which means that the manuscript, images, and Supporting Information files will be freely available online, and any third party is permitted to access, download, copy, distribute, and use these materials in any way, even commercially, with proper attribution. For these reasons, we cannot publish previously copyrighted maps or satellite images created using proprietary data, such as Google software (Google Maps, Street View, and Earth). For more information, see our copyright guidelines: http://journals.plos.org/plosone/s/licenses-and-copyright.

a. You may seek permission from the original copyright holder of Figures 1 and 2 to publish the content specifically under the CC BY 4.0 license. 

Reviewers' comments:

Reviewer's Responses to Questions

**Comments to the Author**

1. Is the manuscript technically sound, and do the data support the conclusions?

Reviewer #1: Yes

Reviewer #2: Yes

Reviewer #3: Yes

Reviewer #4: Yes

2. Has the statistical analysis been performed appropriately and rigorously? 

Reviewer #1: I Don't Know

Reviewer #2: Yes

Reviewer #3: Yes

Reviewer #4: Yes

3. Have the authors made all data underlying the findings in their manuscript fully available?

Reviewer #1: Yes

Reviewer #2: Yes

Reviewer #3: Yes

Reviewer #4: Yes

4. Is the manuscript presented in an intelligible fashion and written in standard English?

Reviewer #1: Yes

Reviewer #2: Yes

Reviewer #3: Yes

Reviewer #4: Yes

5. Review Comments to the Author

Reviewer #1: Abstract:

Lines 31 - 38 should read: in general, the spatial distribution of national agricultural cultural heritage sites during 2013-2021 became increasingly obvious; the center of gravity of spatial distribution showing showings an offset trajectory of ‘north-east, then south-east’. The direction of is in the north-east-south-west spatial distributions direction; including the national-level agricultural cultural heritage sites are rich and diverse, involving farmland landscapes, composite systems, crop varieties, vegetables and melons, tea, forest fruits, special products, farmland irrigation, and animal breeding, in which forest Forest and fruit class heritage site dominate.

Line 43 New sentence: At the county level

Line 47 ②, ③ (??)

Lines 45 - 47 Define LL, LH, HH types

Lines 47 should read In terms of influencing factors, the spatial pattern of national-level agricultural cultural heritage is influenced by temperature, precipitation, level of economic development, population status, transportation accessibility, cultural environment and other factors. Strong interaction influences are evident between these factors and these also affect the spatial distribution of national-level agricultural cultural heritage.

(delete "Strong interaction")

In summary, the spatial pattern of national-level agricultural cultural heritage varies greatly, the spatial distribution is extremely unbalanced, and the regional spatial distribution varies significantly, with strong national and regional characteristics. The development of the agricultural cultural heritage has a strong correlation with the tourism industry as it gradually assumes a distribution of “widely distributed, locally concentrated”.

Line 106- delete “connotation

Lines 114-117 – delete sentence

Line 132 – Academia should read Academic

Lines 141-143 – delete

141 “the selection of indicators has mostly focused on social,

142 economic and natural factors and has lacked consideration of cultural factors. and

143 tourism factors on the spatial layout of agricultural cultural heritage. The”

Line 212 The term “gravity” is not appropriate. Change header to “Centres for standard deviation ellipse analyses”

Line 213 center of gravity shift should read “center of gravity shift” since it a published name but the term”gravity is not appropriate.

Line 215 delete of gravity

Line 216 …according to the change trajectory of the center of gravity…. should read according to the changed trajectory of the center of agricultural cultural heritage sites

Line 293 delete gravity –

Line 301 delete Gravity of

Line 306 center of gravity should read center of agricultural cultural heritage

Line 708 ……interacts and reinforces….. should read . interact and reinforce

Line 748 center-of-gravity should read “center-of-gravity” as noted above

Line 755 center-of-gravity should read “center-of-gravity” as noted above

Reviewer #2: In this paper, spatial analysis methods such as the nearest-neighbor index, center-of-gravity-standard deviation ellipse, kernel density, spatial autocorrelation and geographic probes are used to explore the spatial distribution characteristics and driving factors of national agricultural cultural heritage. Generally, the structure of the paper is clear and complete, but the theoretical depth is general, and there is still room for improvement in the depth and breadth of the research. Here are some comments:

1.Abstract. The description of the study context/importance of the study is relatively good, the study methods and data are clearly described. However, if possible，these parts should be more simply and the main findings part should be more accurate and detailedly.

2.The introduction should outline the background of the undertaken research. However, the research gap is not discussed deeply enough. Increase the summary and brief review of relevant research literature, and highly refine the shortcomings of research and the main innovations or contributions of this manuscript.

3.Figures. The format of the figures should carefully refer to the journal requirements. Some words in the picture cannot be read clearly. Tables. The format of the tables should carefully refer to the journal requirements.

4.The layout of the full text (Including references)is very rough, and it is suggested that the author should carefully revise and improve according to the standards of the journal.

Reviewer #3: Title:

ok.

Abstract: Line 25- Modify abstract start (To…)

Line 52; replace word summary. Go for concluding words or line with a future prospect’s recommendations.

Introduction

Line 125, Check citations format of journal again.

Line 156 re write objective section. Don’t start line with backets

Materials and Methods

Line 164 mention/provide data collection year information.

Line 194 check with mistakes. (start line with capital word “value”)

Results

Presented Well.

Discussion:

• Ok, but include citations for the discussion. Authors can present their results by using examples of other previous studies

Conclusion:

Rephrase heading to conclusion and future prospects.

General comments

After carefully reviewing the manuscript, I must commend the author for their skillful writing and overall presentation. However, I have identified several areas where the manuscript could be improved. These suggestions will help the author further enhance the manuscript's readability, structure, and impact.

Reviewer #4: 1.Some parts of the manuscripts are not written properly, such as L142, L51-52······;

2.The format of the references is not uniform;

3.The driving factors should be considered more agriculture-related,such as farmers' income, agricultural enterprise efficiency and government input to agriculture;

4.Are the factors influencing agricultural heritage the same in different types of agriculture? For example, food crops, cash crops, forestry？

6. PLOS authors have the option to publish the peer review history of their article (what does this mean?). If published, this will include your full peer review and any attached files.

Reviewer #1: No

Reviewer #2: **Yes: **Wentai Bi

Reviewer #3: No

Reviewer #4: No

---

## [Author Response · Author response to Decision Letter 0]

30 Oct 2024

Dear Reviewer，

Thank you very much for your comments. We have carefully studied all your constructive and helpful suggestions and attempted to incorporate them into the current version of the paper, which has undoubtedly greatly improved it. Please refer to the following point-by-point responses to your comments, with our replies and modifications highlighted in blue.

Comments from Reviewer 1

Comment 1:

 Abstract:Lines 31 - 38 should read: in general, the spatial distribution of national agricultural cultural heritage sites during 2013-2021 became increasingly obvious; the center of gravity of spatial distribution showing showings an offset trajectory of ‘north-east, then south-east’. The direction of is in the north-east-south-west spatial distributions direction; including the national-level agricultural cultural heritage sites are rich and diverse, involving farmland landscapes, composite systems, crop varieties, vegetables and melons, tea, forest fruits, special products, farmland irrigation, and animal breeding, in which forest Forest and fruit class heritage site dominate.

Thank you very much for your careful review and valuable suggestions on this article. Based on your feedback, I have made appropriate changes to the summary section to ensure accuracy and fluency in expression. We have revised “in general, the spatial distribution of national agricultural cultural heritage sites… in which forest and fruit class heritage dominates.”to “From an overall perspective, the spatial pattern of China's national agricultural cultural heritage changed greatly from 2013 to 2021, with a highly uneven spatial distribution, gradually showing a distribution pattern of "widely distributed, locally concentrated". The spatial distribution of China's national agricultural cultural heritage is increasingly evident, and the spatial distribution type has evolved from discrete to clustered. The spatial distribution center of gravity shows a shift trajectory of "north-east, then south-east". During the study period, the X axis of the standard deviation ellipse was always greater than the Y axis, but the difference between the X and Y axes was small, indicating that the spatial distribution direction was northeast-southwest, but the directionality was weak. The types of national agricultural cultural heritage are diverse and rich, involving farmland landscapes, composite systems, crop varieties, vegetables and melons, tea, forest fruits, special products, farmland irrigation, and animal breeding, in which forest Forest and fruit class heritage site dominate.”

Comment 2:

Line 43 New sentence: At the county level

We thank you for the comment. We concisely outline the research logic of this paper regarding the new sentence. This paper initially examines the spatial distribution of China's national agricultural cultural heritage from a holistic national perspective, analyzing its central shift trajectory and overall spatial distribution direction. Subsequently, given the localized clustering pattern observed in the spatial distribution of these heritages, which spans multiple cities across China, the paper delves into the high-density clusters of national agricultural cultural heritage from an urban agglomeration perspective. Furthermore, to enhance the scientific rigor and granularity of the study, this paper draws heavily on the extensive research conducted by Chinese scholars in the field of spatial autocorrelation. While previous studies often employed prefecture-level cities or county-level administrative regions as analytical units, the relatively vast geographical scope of prefecture-level cities may not adequately capture the nuanced spatial correlation characteristics of national agricultural cultural heritage. Consequently, this paper selects county-level administrative regions in China as the research unit. This choice aims to refine the analytical granularity, ensuring a more precise exploration of the spatial distribution patterns of agricultural cultural heritage, thereby providing more comprehensive data support and theoretical foundations for conservation and inheritance efforts.

Comment 3:

Line 47 ②, ③ (??)

Thank you for pointing out the problems in our paper. In the abstract section, we have summarized the three key points of the entire article: ① Firstly, we provide a national overview of the centrally shifted trajectory and directional trends in the spatial distribution of China's national agricultural cultural heritage. ② Secondly, from the regional perspectives of urban agglomerations and county-level administrative regions, this paper separately analyzes the agglomeration areas and spatial autocorrelation in the spatial distribution of China's national agricultural cultural heritage. ③ Lastly, in terms of influencing factors, this paper summarizes the extent to which factors such as temperature, precipitation, economic development level, population status, transportation accessibility, and cultural environment influence the spatial distribution of China's national agricultural cultural heritage.

Comment 4:

Lines 45 - 47 Define LL, LH, HH types

We appreciate the suggestion and agree that defining the LL, LH, HH types is important to the reader of the manuscript.The article places the definitions of LL, LH, and HH types on line 477-480.

Comment 5:

Lines 47 should read In terms of influencing factors, the spatial pattern of national-level agricultural cultural heritage is influenced by temperature, precipitation, level of economic development, population status, transportation accessibility, cultural environment and other factors. Strong interaction influences are evident between these factors and these also affect the spatial distribution of national-level agricultural cultural heritage.

(delete "Strong interaction")

In summary, the spatial pattern of national-level agricultural cultural heritage varies greatly, the spatial distribution is extremely unbalanced, and the regional spatial distribution varies significantly, with strong national and regional characteristics. The development of the agricultural cultural heritage has a strong correlation with the tourism industry as it gradually assumes a distribution of “widely distributed, locally concentrated”.

Thank you for your careful revision of our manuscript.We have reorganized and refined the conclusions regarding the influencing factors in the paper to ensure greater accuracy.The revised and improved content is located between lines 58 and 66 of the paper.

Furthermore, after in-depth discussions, we have ultimately decided to refine and integrate the content of the "In summary,…widely distributed, locally concentrated" section into the findings and results.

Comment 6:

Line 106- delete “connotation

We were really sorry for our careless mistakes. Thank you for your re-minder.

Comment 7:

Lines 114-117 – delete sentence

Thank you for your careful revision of our manuscript.

Comment 8:

Line 132 – Academia should read Academic

We feel sorry for our carelessness.in our resubmitted manuscript, the typo is revised. Thanks for your correction.

Comment 9:

Lines 141-143 – delete

141 “the selection of indicators has mostly focused on social,

142 economic and natural factors and has lacked consideration of cultural factors. and

143 tourism factors on the spatial layout of agricultural cultural heritage. The”

Thank you very much for your advice. We strongly agree with your suggestion and have already removed the relevant content.

Comment 10:

Line 212 The term “gravity” is not appropriate. Change header to “Centres for standard deviation ellipse analyses”

We feel great thanks for your professional review work on our article. Following your suggestion. The title was revised and rewritten as follow “Centres for standard deviation ellipse analyses”.

Comment 11:

Line 213 center of gravity shift should read “center of gravity shift” since it a published name but the term”gravity is not appropriate.

Thank you very much for your advice. We have revised "The center of gravity shift model" to "The center shift model"

Comment 12:

Line 215 delete of gravity

Thank you for your careful revision of our manuscript. We have fixed the misrepresentation.

Comment 13:

Line 216 …according to the change trajectory of the center of gravity…. should read according to the changed trajectory of the center of agricultural cultural heritage sites

Thank you for your careful revision of our manuscript. We have revised the original sentence from “according to the change trajectory of the center of gravity” to “according to the changed trajectory of the center of agricultural cultural heritage sites”

Comment 14:

Line 293 delete gravity –

Thanks for your careful checks. We are sorry for our carelessness. Based on your comments, we have made the corrections to make the word harmonized within the whole manuscript.

Comment 15:

Line 301 delete Gravity of

Thank you very much for your advice. the article has fixed the misrepresentation.

Comment 16:

Line 306 center of gravity should read center of agricultural cultural heritage

Thank you for your careful revision of our manuscript. We have revised“center of gravity” to “center of agricultural cultural heritage”.

Comment 17:

Line 708 ……interacts and reinforces….. should read . interact and reinforce

Thank you for your careful revision of our manuscript. We have revised“interacts and reinforces” to “interact and reinforce”.

Comment 18:

Line 748 center-of-gravity should read “center-of-gravity” as noted above

We sincerely thank the reviewer for careful reading.

Comment 19:

Line 755 center-of-gravity should read “center-of-gravity” as noted above

Thank you again for your careful revision of our manuscript.

Comments from Reviewer 2

In this paper, spatial analysis methods such as the nearest-neighbor index, center-of-gravity-standard deviation ellipse, kernel density, spatial autocorrelation and geographic probes are used to explore the spatial distribution characteristics and driving factors of national agricultural cultural heritage. Generally, the structure of the paper is clear and complete, but the theoretical depth is general, and there is still room for improvement in the depth and breadth of the research. Here are some comments:

We feel great thanks for your professional review work on our article.As you are concerned,there are several problems that need to bead dressed.According to your nice suggestions, we have made extensive corrections to our previous draft, the detailed corrections are listed below.

Comment 1:

Abstract. The description of the study context/importance of the study is relatively good, the study methods and data are clearly described. However, if possible，these parts should be more simply and the main findings part should be more accurate and detailedly.

We deeply appreciate your meticulous review of our paper and the insightful suggestions you have provided. Based on your comments, we have revised the background, research methodology, and data description sections of the abstract. Furthermore, we have revised the primary findings to ensure greater precision and comprehensiveness. The adjustments can be found in lines 19 to 60 of the paper.

Comment 2:

The introduction should outline the background of the undertaken research. However, the research gap is not discussed deeply enough. Increase the summary and brief review of relevant research literature, and highly refine the shortcomings of research and the main innovations or contributions of this manuscript.

We sincerely appreciate the valuable comments. We have checked the literature carefully and have added a summary and brief review of relevant research literature in the introduction section, while also conducting an in-depth refinement and overview of the limitations of the study as well as the main innovations or contributions of this paper. The summary and brief review of relevant research literature are located between lines 105 and 124 of the paper. The limitations of the current research are clearly pointed out in lines 139-150. And the main innovations or contributions of this paper are comprehensively and concisely summarized in lines 156-169.

Comment 3:

Figures. The format of the figures should carefully refer to the journal requirements. Some words in the picture cannot be read clearly. Tables. The format of the tables should carefully refer to the journal requirements.

We feel sorry for our carelessness.in our resubmitted manuscript these issues have been revised. Thanks for your correction.

Comment 4:

The layout of the full text (Including references)is very rough, and it is suggested that the author should carefully revise and improve according to the standards of the journal.

We sincerely thank the reviewer for careful reading. We have carefully revised and refined the entire manuscript in accordance with the norms and standards of the journal, aiming to provide readers with a clearer, smoother, and more comprehensible reading experience.

Comments from Reviewer 3

Comment 1:

Title:ok.

We thank you for comment.

Comment 2:

Abstract: Line 25- Modify abstract start (To…) 

We think this is an excellent suggestion. We have further revised the abstract, striving to make the descriptions of the background and methodology more concise while accurately and comprehensively summarizing the research findings of the paper. The revised content is located in lines 19-60 of the paper.

Comment 3:

Line 52; replace word summary. Go for concluding words or line with a future prospect’s recommendations.

Thank you again for your careful revision of our manuscript. We strongly agree with your viewpoint. After reviewing relevant literature, we have also come to realize that the word "summary" is not appropriate to use here. However, after in-depth discussion, we believe that placing these contents in the conclusion or findings section would be more suitable.

Comment 4:

Introduction

Line 125, Check citations format of journal again.

Thanks for your correction. We have revised the format of the paper according to the journal's format requirements. 

Comment 5:

Line 156 rewrite objective section. Don’t start line with backets

Materials and Methods

 We sincerely appreciate the valuable comments. We have rewritten this section, and it is located in lines 156-169 of the paper.

Comment 6:

Line 164 mention/provide data collection year information.

Thank you for your suggestion.We have added the year information of data collection in line 174 of the paper.

Comment 7:

Line 194 check with mistakes. (start line with capital word “value”)

We feel sorry for our carelessness.in our resubmitted manuscript the typo is revised. Thanks for your correction.

Comment 8:

Results

Presented Well.

We thank you for comment.

Comment 9:

Discussion:

Ok, but include citations for the discussion. Authors can present their results by using examples of other previous studies

We sincerely appreciate the valuable comments. In the discussion section, we have cited the results of previous studies as examples, with the relevant content located in lines 730-736 of the paper.

Comment 10:

Conclusion:

Rephrase heading to conclusion and future prospects.

We think this is an excellent suggestion. Based on your suggestion, we have changed "Conclusion " to "Conclusion and Future Prospects ".

Comment 11:

General comments

After carefully reviewing the manuscript, I must commend the author for their skillful writing and overall presentation. However, I have identified several areas where the manuscript could be improved. These suggestions will help the author further enhance the manuscript's readability, structure, and impact.

Thank you again for your careful revision of our manuscript.

Comments from Reviewer 4

Comment 1:

Some parts of the manuscripts are not written properly, such as L142, L51-52······;

Thank you for your valuable suggestions. We have made corresponding adjustments to these parts of the article to facilitate better reading and understanding for the authors.

Comment 2:

The format of the references is not uniform;

Thanks for your careful check. We are sorry for our carelessness. We have re

---

## [Editor Report · Decision Letter 1]

4 Nov 2024

Study on the Evolution of the Spatial Patterns and Driving Factors of National Agricultural Cultural Heritage in China

PONE-D-24-22604R1

Dear Dr. LIU,

We’re pleased to inform you that your manuscript has been judged scientifically suitable for publication and will be formally accepted for publication once it meets all outstanding technical requirements.

Kind regards,

Bijay Halder

Academic Editor

PLOS ONE
---

## [Editor Report · Acceptance letter]

7 Nov 2024

PONE-D-24-22604R1 

PLOS ONE

Dear Dr. LIU, 

I'm pleased to inform you that your manuscript has been deemed suitable for publication in PLOS ONE. Congratulations! Your manuscript is now being handed over to our production team.

Kind regards, 

on behalf of

Mr. Bijay Halder 

Academic Editor

PLOS ONE